# On-Demand Unlabeled Personalized Federated Learning

## Abstract

In *Federated Learning* (FL), multiple clients collaborate to learn a shared model through a central server while keeping data decentralized. Personalized Federated Learning (PFL) further extends FL by learning a personalized model per client. In both FL and PFL, all clients participate in the training process and their labeled data are used for training. However, in reality, novel clients may wish to join a prediction service **after it has been deployed**, obtaining predictions for their own **unlabeled** data.

Here, we introduce a new learning setup, *On-Demand Unlabeled PFL* (OD-PFL), where a system trained on a set of clients, needs to be later applied to novel unlabeled clients at inference time. We propose a novel approach to this problem, ODPFL-HN, which learns to produce a new model for the late-to-the-party client. Specifically, we train an encoder network that learns a representation for a client given its unlabeled data. That client representation is fed to a hypernetwork that generates a personalized model for that client. Evaluated on five benchmark datasets, we find that ODPFL-HN generalizes better than the current FL and PFL methods, especially when the novel client has a large shift from training clients. We also analyzed the generalization error for novel clients, and showed analytically and experimentally how novel clients can apply differential privacy to protect their data.

## 1 Introduction

Federated Learning (FL) is the task of learning a model over multiple disjoint local datasets, while keeping data decentralized (25). Personalized Federated Learning (PFL) (38) extends FL to the case where the data distribution varies across clients. PFL has numerous applications from a smartphone application that wishes to improve text prediction without uploading user-sensitive data, to a consortium of hospitals that wish to train a joint model while preserving the privacy of their patients. Current PFL methods assume that all clients participate in training and that their data is labeled , so once a model is trained, a novel client cannot be added.

In many cases, however, a federated model has been trained and deployed, but then novel clients wish to join. Often, such novel clients do not have labeled data, and their data distribution may shift from that of training clients. This is the case, for example, when a speech recognition federated model has been deployed and needs to be applied to new users or when a virus diagnostic has been developed for some regions or countries, and then needs to be applied to new populations while the virus spreads. This learning setup poses a hard challenge to existing approaches. Non-personalized FL techniques may not generalize well to novel clients due to domain shift. PFL techniques learn personalized models but are not designed to be applied to a client that was not available during training.

When a novel client joins with its *labeled* data, there are various strategies to adapt the pre-trained model to the novel client. For instance, a FL model can be fine-tuned using those labels. For PFL, it is less clear which personalized model should be fine-tuned. (29) used a hypernetwork (HN) that generates a personalized model for each client, given a descriptor of client labels. To generalize to a novel client with labeled data, they fine-tuning the descriptor using the labeled data through the hypernetwork. While all these approaches are useful, they cannot handle novel clients that have no labeled data.

Here, we define a novel problem: performing federated learning on novel clients with unlabeled data that are only available at inference time. We call this setup **OD-PFL** for *On-Demand Unlabeled Personalized*

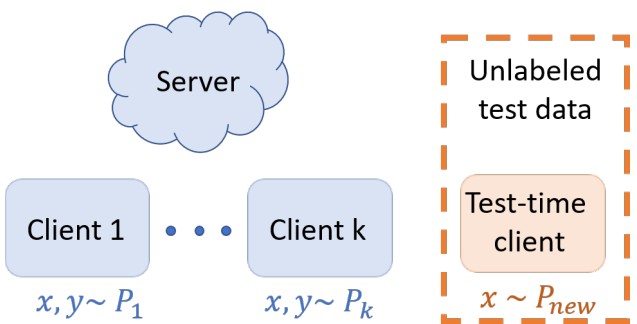

Figure 1: **The On-demand PFL problem.** A set of $k$ clients $c_1, \ldots, c_k$ are available for training a PFL model, each with their own distribution $P_i(x, y)$. After training is completed, a new client appears, with its own distribution $P_{new}$ over unlabeled data. The goal is to create a model $f_{new}$ that minimizes the loss over the new client data $l(y, f_{new}(x))$.

*Federated Learning.* We propose a novel approach to this problem, called ODPFL-HN. During training, our architecture learns a space of personalized models, one for each client, together with an encoder that maps each client to a point in that client space. All personalized models are learned jointly through an HN, allowing us to combine personalized data effectively. At inference time, a novel client can locally compute its own descriptor using the client encoder. Then, it sends the descriptor to the server as input to the HN and obtains its personalized model.

A key question remains for this approach to succeed: How to compute a descriptor of a novel unlabeled client? A key idea is to define a *client encoder* that maps a dataset into a descriptor, and train it jointly with the HN. We explore the properties that this encoder should have. First, its architecture should implement a function that is invariant to permutations over its inputs (35). Furthermore, if objects in the data adhere to their own symmetries, such as images, graphs, or point clouds, the encoder architecture should also be invariant to these symmetries (24). To the best of our knowledge, this is the first paper that discusses learning such invariant descriptors of datasets or clients.

FL is motivated by privacy, but was shown to be vulnerable (21). In the OD-PFL setup, novel clients do not expose data or gradients, so they can better control their privacy. We show theoretically how differential privacy (DP) can be applied effectively to a new client and then experimentally measured how DP affects the accuracy of the personalized model.

This paper makes the following contributions: (1) A new learning setup, OD-PFL, learning a personalized model to **novel unlabeled clients at inference time**. (2) A new approach, learn a space of models using an encoder that maps an unlabeled client to that space, and an architecture ODPFL-HN based on hypernetworks. (3) A generalization bound based on multitask learning and domain adaptation, and analysis of differential privacy for a novel client. (4) Evaluation on five benchmark datasets, showing that ODPFL-HN performs better than or equal to the baselines.

## 2   Related work

**Federated learning (FL)**. In FL, clients collaboratively solve a learning task, while preserving data privacy and maintaining communication efficiency. By collaborating through FL, clients leverage the shared pool of knowledge from other clients in the federation, and can better handle data scarcity, low data quality, and unseen classes. The literature on FL is vast and cannot be covered here. We refer the reader to recent surveys (1; 36). We note that (20) addresses clients with unlabeled data but does not address novel (test-time) clients, which is the main focus of this paper.

**Personalized Federated Learning (PFL)**. FL methods learn a single global model across clients, and this limits their ability to deal with heterogeneous clients. In contrast, PFL methods are designed to handle heterogeneity of data between clients by learning multiple models. (30) separates PFL methods into data-

based and model-based approaches. Data-based PFL approaches aim to smooth the statistical heterogeneity of data among different clients, by normalizing the data (7) or by selecting a subset of clients with minimal class imbalance (32). Model-based PFL approaches adapt to the diversity of data distributions across clients. As an example, a server may learn a global model and share it with all the clients. Then, each client learns its own local model on top of the global model. We refer the reader to recent surveys (16; 30).

**Novel labeled clients**. Several recent studies proposed to create personalized models for a novel **labeled** client. (29) used an HN to produce personalized models for training clients. Given a novel client, they tuned the embedding layer of the HN using client labels. In (22), each client interpolates a global model and a local kNN model is produced using the labels of the novel clients. (23) modeled each client as a mixture of distributions using an EM-like algorithm. A new labeled client uses its labels to calculate its own personalized mixture. All these methods depend on having access to labels of the novel client hance do not apply to the OD-PFL setup. In Sec. 6.2, we test 3 different ways to apply existing PFL models to our setup and show that ODPFL-HN outperforms all these variants.

Adapting a model to a new distribution during inference can be viewed as a variant of domain adaptation (DA) (31; 14; 19). Our on-demand setup can be viewed as an adaptation of DA to FL. We emphasize that DA is fundamentally different from FL, since in FL the data is distributed across different clients. As a result, DA approaches cannot be applied directly to the FL setup. To the best of our knowledge, this is the first paper to address test-time adaptation in a FL setup.

**Differential privacy (DP)**. The goal of DP is to share information about a dataset but to avoid sharing information about individual samples in the dataset. Although privacy is a key motivation of PFL, private information is exposed in the process: an adversarial client can infer the presence of exact data points in the data of other clients, and under certain conditions even generate the data of other clients. See a survey for more details (26). A natural solution is to use DP to protect client privacy (10). However, DP adds noise to the training process and may harm the model performance. Here, we focus on the privacy of a novel client at test time, so the trained models remain untouched, and each novel client can choose its own privacy-accuracy trade-off at inference time.

## 3   The Learning Setup

We now formally define the learning setup of OD-PFL. We follow the notation in (2). Let $X$ be an input space and $Y$ an output space. $P$ is a probability distribution over the data $X \times Y$. Let $l$ be a loss function $l : Y \times Y \to R$. $\mathcal{H}$ is a set of hypotheses with $h : X \to Y$. The error of a hypothesis $h$ over a distribution $P$ is defined by $err_P(h) = \int_{X \times Y} l(h(x), y) dP(x, y)$.

In OD-PFL, we train a federation of $N$ clients $c_1, \ldots, c_N$, and other, novel, clients are added at inference time. For simplicity of notation, we consider a single novel client $c_{new}$. Let $\{P_i\}_{i=1}^{N}$ be the data distributions of training clients, and $P_{new}$ the data distribution of the novel client $c_{new}$. Each training client has access to $m_i$ IID samples from its distribution $P_i$, denoted as $S_i = \{(x_j^i, y_j^i)\}_{j=1}^{m_i}$.

The goal of OD-PFL is to use data from training clients $\{S_i\}_{i=1}^{N}$ to learn a mechanism that can assign a hypothesis $h_{new} \in \mathcal{H}$ when given unlabeled data from a novel client $S_{new} = \{x_j^{new}\}_{j=1}^{m}$. This hypothesis should minimize the expected error of the novel client. For any distribution of its data $P_{new}$, that error is defined by $err_{P_{new}}(h_{new}) = \int_{X \times Y} l(h_{new}(x), y) dP_{new}(x, y)$.

## 4   Our approach

We introduce OD-PFL to address the challenge of producing an "on-demand" personalized model for new unlabeled clients after a federated model has been deployed. This task is difficult because the data distribution of a novel client may differ from that of training clients and is unknown at training time. Also, the training clients are no longer available at inference time. To the best of our knowledge, these constraints were not considered in the FL setup before, and existing methods are not designed to handle this new setup. Generalizing to a novel client is hard for FL methods, because they do not specialize. It is hard for PFL methods, because no single model learned during training would not necessary fit a novel client. Even if it

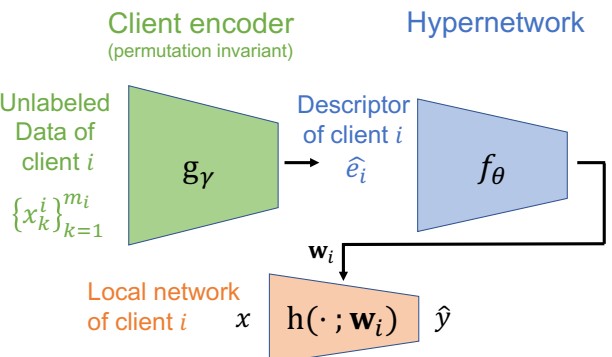

Figure 2: **The high-level architecture structure of ODPFL-HN.** Given a client $c_i$ with $m_i$ unlabeled samples $x_1, \ldots, x_{m_i}$, the client encoder $g_\gamma$ produces an embedding $\hat{e}_i$. Then, the hypernetwork $f_\theta$ predicts weights $w_i$ for the local model $h_i$ of client $i$.

did, there is no clear way to select that model, because the novel client has no labels to use as a selection criteria. Here, we propose a meta-learning mechanism to produce a model that fits the new distribution of unlabeled data.

A natural candidate for such a meta-mechanism would be hypernetworks (HNs). HNs are neural networks that output the weights of another network and can therefore be used to produce "on-demand" models. Since the weights of the generated model are a (differentiable) function of the HN parameters, training the HN is achieved simply by propagating gradients from the generated (client) model. To generate a model for a client, the HN should be fed with a descriptor that summarizes the client dataset. Here we propose to learn a *client encoder* that takes as input the unlabeled data of a client and produces a dense descriptor. Figure 2 illustrates our approach. Each client feeds its input samples to a client encoder that produces an embedding vector. Then an HN takes the embedding and produces a personalized model. During training, the client uses its labels to tune the personalized model and back-propagates the gradients to the HN and the client encoder. We now describe our approach in detail.

**Client encoding.** The goal of client encoding is to map an entire data set into a dense descriptor. Formally, it maps unlabeled samples $S_i = \{x_j^i\}_{j=1}^{m_i}$ of client $i$ to a descriptor $e_i$ embedded in a representation space $\mathcal{E}$. In essence, the encoder is expected to map similar datasets to nearby descriptors in a way that balances personalization – of unique clients, with generalization – across similar clients.

For this mapping to be effective, it should obey several properties. First, the embedding should be inductive and generalizing, in the sense that the embedding function would later be applied to a novel client at test time and should generalize to that client. Second, since data samples form an unordered set, we wish that the encoder obeys the set symmetries of the data and is invariant to permutations over samples. Third, the encoder should capture the full data set. We now discuss our design decisions when building the encoder.

First, with respect to generalization. One may be tempted to explicitly regularize the representation such that similar datasets are mapped to close vectors in the embedding space. However, note that the descriptors are consumed by the downstream HN. Therefore, training the two networks jointly, while regularizing their parameters, should yield a representation that generalizes across clients. This is because the encoder tunes the representation to fit the downstream HN.

Second, with respect to permutation invariance, we tested three architectures. (1) *DeepSet* (DS) (35). In DS, each data point is fed to the same ("siamese") model and produces a feature vector. Then, an invariant pooling operator (usually mean) is applied to all outputs, and then processed by a second model, yielding the final descriptor. (2) DS is invariant to permutations over input samples, but it does not take into account symmetries of each element itself, such as translation invariance in images. We use *Deep Sets for symmetric elements* (DSS) (24) to handle symmetries at both the set level and the element level. (3) DS and DSS uniformly aggregate information from all individual elements in the set. Sometimes, it is beneficial to

---

**Algorithm 1** Training On-demand PFL-HN

**Server executes:**
    initialize $\theta, \gamma$
    **for** each round $t = 1, 2, ...$ **do**
        $i \leftarrow$ select a random client out of $N$ clients
        $e_i \leftarrow$ CLIENTENCODING$(g_\gamma)$
        $w_i \leftarrow f_\theta(e_i)$                                           ▷ server computes a personal model
        $\Delta w_i \leftarrow$ CLIENTUPDATE$(w_i)$
        apply chain rule to obtain $\Delta\theta$ and $\Delta e_i$ from $\Delta w_i$
        $\theta \leftarrow \theta - \eta\Delta\theta$
        $\Delta\gamma \leftarrow$ CLIENTBACKPROP$(\Delta e_i)$
        $\gamma \leftarrow \gamma - \eta\Delta\gamma$

**function ClientEncoding**$(g_\gamma)$ //Run on client $i$
    $e_i \leftarrow g_\gamma(\{x_k\}_{k=1}^{m_i})$                            ▷ client $i$ computes its embedding
    **return** $e_i$ to server

**function ClientUpdate**$(w_i)$ //Run on client $i$
    $\mathcal{B} \leftarrow$ split $\{x_k\}_{k=1}^{m_i}$ into batches
    $w_i^{new} \leftarrow w_i$
    **for** each local epoch $e$ from 1 to $E$ **do**
        **for** each batch $b \in \mathcal{B}$ **do**
            update $w_i^{new}$ using $l(h(\cdot; w_i^{new}); b)$                  ▷ training
    $\Delta w_i \leftarrow w_i^{new} - w_i$
    **return** $\Delta w_i$ to server

**function ClientBackprop**$(\Delta e_i)$ //Run on client $i$
    apply chain rule to obtain $\Delta\gamma$ from $\Delta e_i$
    **return** $\Delta\gamma$ to server

---

consider several elements together when computing the descriptor. To capture sample-to-sample interactions, we used a *set transformer* (ST) (18). ST uses attention to aggregate representations of all elements into a single descriptor. The weight of each element is determined by the context of other elements in the dataset. We treated the architecture as a hyperparameter and selected it using the validation set.

Finally, with respect to describing the full dataset. The simplest approach is to use large batches that contain the entire dataset as input to the client encoder. We also tested an alternative approach that can be applied to large datasets that do not fit in a single batch in memory. In these cases, we split the data into smaller batches, encoded each batch, and used the average over batch descriptors as the final descriptor. Note that this resembles a DeepSet architecture and that when batches are sampled uniformly at random, would obey in expectation invariance to input permutation.

**Hypernetworks.** Our goal is to create a personalized model for a new client at inference time. Assuming that the client encoder summarized all relevant information to create such a personalized model. A natural solution is to learn a mapping from such descriptors to personalized models, and apply it to the descriptor of a novel client. This is exactly what HNs are designed for.

An HN $f_\theta$ parameterized by $\theta$ embodies a mapping from a client-embedding space to a hypotheses space $f_\theta : \mathcal{E} \to H$. A client with an embedding vector $e_i$ is mapped by the HN to a personalized model $h_i = h(\cdot; w_i)$, with $w_i = f_\theta(e_i)$.

**Training.** The client encoder and the HN are trained jointly. They produce a client descriptor and a personalized model for every client by optimizing the following loss

$$L(\theta, \gamma) = \sum_{i=1}^{n} \sum_{j=1}^{m_i} l\left( f_\theta \Big( g_\gamma(\{x_j^i\}_{j=1}^{m_i}) \Big) \Big( x_j^i \Big), y_j^i \right) \tag{1}$$

using training clients (labeled) $c_1, \ldots, c_N$, where $l$ is a cross-entropy loss.

**Workflow.** Algorithm 1 shows the workflow of ODPFL-HN. During training, in each communication step, we repeat these 4 steps: (1) The server selects a random client and sends to it the current encoder $g_\gamma$. (2) The client locally predicts its embedding $e_i$ and sends it back to the server. (3) Using its embedding, the server uses an HN $f_\theta$ to generate a customized network $h_i = h(\cdot; w_i)$ and communicates it to the client. (4) The client then locally trains that network on its data and communicates back to the server the delta between the weights before and after training. Using the chain rule, the server can train the hypernetwork and the encoder.

At inference time, (1) the server sends the encoder to the novel client. (2) The client uses the encoder to calculate an embedding $e_{new}$ and sends it to the server. (3) The server uses the HN to predict the personalized model of the client $h_{new} = h(\cdot; w_{new})$ from the embedding and sends the result to the client. The client then applies its personalized model locally without revealing its data.

## 5 Generalization bound

The data of the new client and the clients of the federation may be sampled from different distributions. In the general case, there is no guarantee that learning a model for labeled clients would lead to a good model for a novel client. We now show that under reasonable assumptions, previous bounds developed for multitask learning (MTL) and for domain adaptation (DA), can be applied to the OD-PFL setup, to bound the generalization error of the novel client.

Intuitively, the bound has two terms; one captures the domain shift, and the other captures the generalization error.

**Theorem 1.** *Let $\mathcal{H}$ be a hypothesis space, $P_{new}$ be a data distribution of a novel client, and $Q$ be a distribution over the distributions of the clients, that is, $P_i$ is drawn from $Q$. $\hat{err}_z(H)$ is the empirical loss over the training-client data and is defined in detail in the Appendix **??**.*

*The generalization error of a novel client is bounded by $err_{P_{new}}(H) \leq \hat{err}_z(H) + \epsilon + \frac{1}{2} \int_P \inf_{h \in H} \hat{d}_{H\Delta H}(P, P_{new}) dQ(P)$. Here, $\epsilon$ is an approximation error of a client in the federation from Theorem 2 in (2) and $\hat{d}_{H\Delta H}$ is a distance measure between the probability distributions defined in (3).*

*Proof.* See Appendix **??** for a detailed proof.

$\square$

## 6 Experiments

### 6.1 Experiment setup and evaluation protocol

We evaluated ODPFL-HN using five benchmarks using an experimental protocol designed for the OD-PFL setup where novel clients are presented to the server during inference.

**Client split:** To quantify the performance of novel clients, we first randomly partition the clients to $N_{train}$ train clients and $N_{novel}$ novel clients. We use $N_{novel} = N/10$. Unless stated otherwise, we report average accuracy for the novel clients. To conduct a fair comparison, training is limited to 500 steps for all evaluated methods. In each step, the server communicates with a 0.1 fraction of training clients following the protocol of each method.

Table 1: **Accuracy on novel unlabeled clients, CIFAR10 & CIFAR100:** Values are averages across clients and standard error of the means.

| | CIFAR-10 | | | | CIFAR-100 | | | |
|---|---|---|---|---|---|---|---|---|
| split | pathological | $\alpha = 0.1$ | $\alpha = 1$ | $\alpha = 10$ | pathological | $\alpha = 0.1$ | $\alpha = 1$ | $\alpha = 10$ |
| FedAvg | $50.3 \pm 2.9$ | $58.7 \pm 3.6$ | $48.4 \pm 2.9$ | $66.2 \pm 0.5$ | $16.2 \pm 1.3$ | $17.9 \pm 1.0$ | $13.5 \pm 1.7$ | $30.2 \pm 0.4$ |
| FedProx | $54.2 \pm 2.0$ | $53.9 \pm 2.2$ | $54.2 \pm 1.0$ | $52.8 \pm 0.6$ | $5.5 \pm 1.2$ | $15.9 \pm 0.7$ | $20.6 \pm 0.6$ | $12.4 \pm 0.6$ |
| FedMA | $42.9 \pm 1.8$ | $49.3 \pm 3.4$ | $54.5 \pm 0.9$ | $53.8 \pm 0.6$ | $11.2 \pm 0.7$ | $12.6 \pm 1.0$ | $6.5 \pm 0.4$ | $7.3 \pm 0.3$ |
| PFL-sampled | $24.8 \pm 1.0$ | $61.0 \pm 3.7$ | $49.4 \pm 3.0$ | $\mathbf{68.5 \pm 0.7}$ | $3.9 \pm 0.4$ | $13.5 \pm 0.5$ | $3.4 \pm 1.4$ | $32.4 \pm 0.1$ |
| PFL-nearest | $24.4 \pm 6.2$ | $63.1 \pm 3.5$ | $49.4 \pm 0.9$ | $\mathbf{68.5 \pm 0.7}$ | $6.5 \pm 2.7$ | $14.3 \pm 0.6$ | $3.4 \pm 0.4$ | $32.1 \pm 0.2$ |
| PFL-ensemble | $47.6 \pm 3.2$ | $62.2 \pm 3.7$ | $49.4 \pm 3.0$ | $\mathbf{68.5 \pm 0.7}$ | $7.8 \pm 1.8$ | $20.4 \pm 1.2$ | $3.4 \pm 1.4$ | $32.7 \pm 0.2$ |
| ODPFL-HN (ours) | $\mathbf{59.5 \pm 3.5}$ | $\mathbf{66.0 \pm 3.0}$ | $\mathbf{62.9 \pm 1.0}$ | $68.1 \pm 0.5$ | $\mathbf{19.5 \pm 2.1}$ | $\mathbf{26.4 \pm 0.1}$ | $\mathbf{32.9 \pm 0.9}$ | $\mathbf{33.6 \pm 0.1}$ |

**Sample split and HP tuning:** We split the samples of each *training client* into a training set and a validation set. Validation samples were used for hyperparameter tuning. See Appendix **??** for more details.

## 6.2 Baselines

We evaluate using FL methods, which train a single global model, and PFL methods, which train one model per client. We compare the following FL methods: **(1) FedAVG** (25), where the parameters of local models are averaged with weights proportional to the sizes of the client datasets. **(2) FedProx** (27) adds a proximal term to the client cost functions, thereby limiting the impact of local updates by keeping them close to the global model. **(3) FedMA** (33) constructs the shared global model in a layer-wise manner by matching and averaging hidden elements with similar feature extraction signatures. For inference with FL methods, all novel clients are evaluated using the single global model.

Applying PFL methods to OD-PFL is not straightforward because **PFL methods are not designed to generalize to a novel unlabeled client.** They produce a model per training client, but it is not clear how to use these models for inference over a novel client. We tested three different ways to use PFL for inference with a novel client: **(4) PFL-sampled:** Draw a trained client model uniformly at random. We evaluate this baseline by computing the mean accuracy of all personalized models on each novel client. **(5) PFL-nearest:** We used the training client model closest to the novel client. We measure the distance using A-distance (4), which can be calculated in a FL setup. **(6) PFL-Ensemble:** (4) and (5) use a model from one of the training clients as the new client model. Here, we try a stronger baseline that uses all personalized models for a single novel client, by averaging the logits of all models for each prediction. In practice, this method is expensive in communication and computation costs. All experiments used pFedHN (29) to produce models for training clients.

## 6.3 Results for CIFAR

We evaluate ODPFL-HN using CIFAR10 and CIFAR100 (15) using two protocols that were previously suggested in the literature.

**(1) Pathological split:** As proposed by (25), we sort the training samples by their labels and partition them into $N \cdot K$ shards. Then each client is randomly assigned $K$ of the shards. This results in $N$ clients with the same number of training samples and a different distribution over labels. In our experiments, we use $N = 100$ clients, $K = 2$ for CIFAR10 and $K = 5$ for CIFAR100.

**(2) Dirichlet allocation:** We follow the procedure by (12) to control the magnitude of the distribution shift between clients. For each client $i$, samples are drawn independently with class labels following a categorical distribution over classes with a parameter $q_i \sim Dir(\alpha)$. Here, $Dir$ is the symmetric Dirichlet distribution. We conduct three experiments for each of the two datasets with $\alpha \in 0.1, 1, 10$. Smaller values of alpha imply larger distribution shifts between clients.

Table 2: **Accuracy on novel unlabeled clients for iNaturalist, Landmarks and Yahoo Answers.**
Values are averages and SEMs across novel clients.

| | iNaturalist | | Landmarks | Yahoo Answers |
|---|---|---|---|---|
| split | Geo-300 | Geo-1k | User-160k | User-1K |
| FedAvg | $36.1 \pm 1.6$ | $36.9 \pm 1.1$ | $34.8 \pm 1.3$ | $27.6 \pm 0.1$ |
| FedProx | $17.4 \pm 0.8$ | $26.5 \pm 1.7$ | $13.8 \pm 1.0$ | $13.9 \pm 0.1$ |
| FedMA | $13.4 \pm 0.7$ | $17.5 \pm 0.8$ | $3.80 \pm 0.4$ | $25.0 \pm 0.1$ |
| PFL sampled | $25.6 \pm 1.4$ | $27.2 \pm 0.9$ | $37.4 \pm 1.3$ | $33.2 \pm 0.1$ |
| PFL nearest | $24.2 \pm 3.7$ | $26.9 \pm 4.6$ | $33.1 \pm 3.6$ | $13.2 \pm 0.1$ |
| PFL ensemble | $31.5 \pm 1.6$ | $36.6 \pm 1.2$ | $\mathbf{39.1 \pm 1.4}$ | $33.2 \pm 0.1$ |
| ODPFL-HN | $\mathbf{37.5 \pm 1.7}$ | $\mathbf{41.6 \pm 1.2}$ | $\mathbf{41.1 \pm 1.4}$ | $\mathbf{35.8 \pm 0.2}$ |

**Implementation details:** There are three different models in ODPFL-HN: A target model, an HN, and a client encoder. **Target model:** We use a LeNet (17) with two convolutions and two fully connected layers. To assure a fair comparison, we use the same target model across all evaluated methods and baselines. **Client encoder:** For the DS encoder, we use the same architecture as the target model, with an additional fully connected layer followed by pooling operations over batch dimension. These layers are added after each convolution layer and before the fully connected layers. For the DSS and the ST encoder we use the public implementation provided by the authors. **Hypernetwork:** The HN is a fully connected network, with 3 hidden layers and linear head for each target weight tensor.

**Results:** Table 1 compares ODPFL-HN to baselines on CIFAR-10 and CIFAR-100. ODPFL-HN performs better than all baselines in all the evaluated scenarios, except for CIFAR-10 with the $\alpha = 10$ split.

### 6.4 Results for iNaturalist

iNaturalist is a dataset for Natural Species Classification based on the iNaturalist 2017 Challenge (11). The dataset has 1,203 classes. Following (13), we evaluate ODPFL-HN using two geographical splits of iNaturalist: iNaturalist-Geo-1k with 368 clients, and iNaturalist-Geo-300 with 1,208 clients.

**Implementation details:** We use a MobileNetV2 (28) pre-trained on ImageNet to extract features for each image. The extracted feature vectors, of length 1280, are the input for both the target model and the client encoder. **Target model:** The target model is a fully connected network with two Dense layers and a Dropout layer. **Client encoder:** The client encoder has three fully connected layers with pooling operations after the first layer. HN implementation is the same as in Sec. 6.3.

**Results:** Table 2 shows ODPFL-HN outperforms current FL methods and adapted PFL methods.

### 6.5 Results for Landmarks

Landmarks is based on the 2019 Landmark-Recognition Challenge (34). Following (13), we divide the dataset into clients by authorship. The resulting Landmarks-User-160k split contains 1,262 clients with 2,028 classes. Implementation is as in Sec. 6.4.

**Results:** Table 2 shows that ODPFL-HN outperforms current FL methods in all evaluations. ODPFL-HN outperforms the adapted PFL methods with the exception of PFL-ensemble where it ties. However, PFL-ensemble suffers from relatively large communication and computation costs compared to the proposed ODPFL-HN.

### 6.6 Results for Yahoo Answers

Yahoo Answers is a question classification dataset (37), with 1.4 million training samples from 10 classes. We divide the data into 1000 clients using the Dirichlet allocation procedure with $\alpha = 10$.

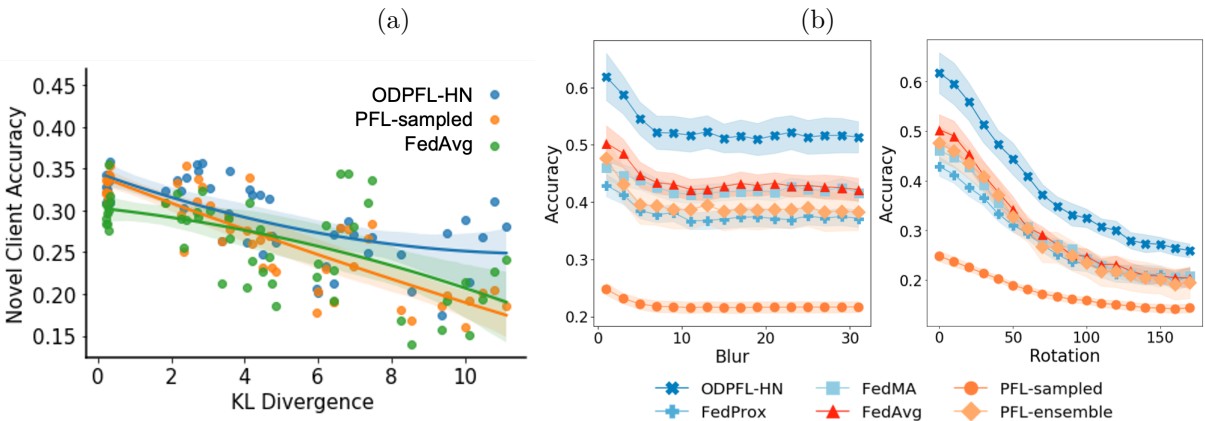

Figure 3: **Distribution shift. (a)** Accuracy of novel clients vs. distribution shift between the novel client and training clients. Shown results for CIFAR-100 across multiple splits to $N_{train} = 90$ train clients and $N_{novel} = 10$ novel clients using symmetric Dirichlet distributions with varying parameter $\alpha$ (see Sec. 6.7). Accuracies of novel clients are reported against the KL-divergence (over label distribution) from the nearest train client for each method. **(b)** Test mean accuracy ($\pm$ S.E.M. over 10 clients) for CIFAR-10 novel clients corrupted with blur using various size of Gaussian filters and rotation at various angles.

**Implementation details:** We used BERT (6) to extract a 768-dimension feature vector for each sample. For more details, see Section 6.4.

**Results:** Table 2 shows that ODPFL-HN outperforms current FL methods and adapted PFL methods.

### 6.7 How distribution shift affects generalization

We expect ODPFL-HN generalization to depend on the similarity between the novel client and the training clients. Novel clients that differ from training clients may perform poorly compared to clients that are similar to training clients.

To quantify this effect, we generated clients at varying similarity levels by creating new splits of CIFAR-100 using Dirichlet allocation while varying $\alpha \in \{0.1, 0.25, 0.5, 1, 10\}$. To measure the similarity between a novel client and all training clients, we computed the empirical label distributions of each client and computed their KL-divergence from the novel client. Figure 3a presents the accuracy of a novel client as a function of its $D_{KL}$ to the nearest train client. As expected, the accuracy decreases as $D_{KL}$ grows, for all evaluated methods. ODPFL-HN demonstrates the most moderate decrease, achieving the highest accuracy in large distribution shifts.

### 6.8 Robustness to covariate shift

Common benchmarks for PFL assume that different clients have a different distribution of labels. Here, we conduct an additional experiment to measure the effect of covariate shifts between clients. We evaluated the robustness of ODPFL-HN to a wide range of blur and rotation corruptions, using the CIFAR10 dataset. We applied the corruptions to the data of the novel client, while the data of training clients was kept uncorrupted. Figure 3b compares the accuracy for novel clients with varying levels of blur and rotation corruption. ODPFL-HN consistently out-performs all baselines.

We test ODPFL-HN robustness to covariate shift, by training an FL model on CIFAR10 (pathological split) and used STL10 as the data of the novel client. ODPFL-HN achieves $43.1\% \pm 4.1\%$, much better than all other baselines: FedProx $35.3\% \pm 2.7\%$, FedMA $32.9\% \pm 2.5\%$, FedAvg $34.7\% \pm 4.1\%$, PFL-sampled $17.6\% \pm 0.6\%$, PFL-ensemble $28.1\% \pm 1.9\%$ and PFL-nearest $31.9\% \pm 3.8\%$.

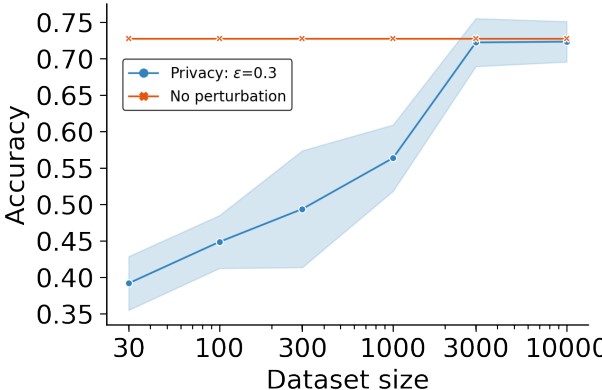

Figure 4: How much data is needed to achieve differential privacy? Test accuracy (±SEM) for CIFAR-10 novel client as a function of datasets size. The blue curve depicts models with privacy level of $\epsilon = 0.3$. With more data, smaller perturbation is needed to achieve a desired level of privacy, as a result, the accuracy rises when more data is used.

# 7 Differential Privacy

A key aspect of FL is data privacy, since it does not require clients to share their data directly with the hub. Unfortunately, some private information may be exposed (see a recent survey by 26). In this section, we analyze the privacy of a novel client and characterize how it can protect its privacy by applying differential privacy (DP) (9). We further study the trade-off between the privacy and accuracy of personalized models.

We first define key concepts and our notation. We use $(\epsilon, \delta)$-DP as defined by (8). Two datasets $D, D'$ are *adjacent* if they differ in a single instance.

**Definition 7.1.** A randomization mechanism $M : D \to R$ satisfies a $(\epsilon, \delta)$-differential privacy if for any two adjacent inputs $d, d' \in D$ and for any subset of outputs $S \subseteq R$ it holds that $Pr[M(d) \in S] \leq e^\epsilon Pr[M(d') \in S] + \delta$.

Here, $\epsilon$ quantifies privacy loss, where smaller values mean better privacy protection, and $\delta$ bounds the probability of privacy breach. The *sensitivity* of a function $f$ is defined by $\Delta f = \max_{D,D'} ||f(D) - f(D')||$, for two datasets $D$ and $D'$ that differ by only one element.

It was shown in (8) that given a model $f$, data privacy can be preserved by perturbing the output of the model and calibrating the standard deviation of the perturbation according to the sensitivity of the function $f$ and the desired level of privacy $\epsilon$. Intuitively, if the protected model is not very sensitive to changes in a single training element, one can achieve DP with smaller perturbations.

Our focus here is to apply DP to a novel client that joins a pre-trained ODPFL-HN model. Fortunately, since the novel client does not participate in training, the only information that a novel client shares with the server is the client descriptor. This descriptor is computed locally by the client, so applying DP to the encoder can protect its data privacy.

Several mechanisms were proposed to achieve DP by adding noise. (8) describes a Gaussian mechanism that adds noise drawn from a Gaussian distribution with $\sigma^2 = \frac{2\Delta f^2 \log(1.25/\delta)}{\epsilon^2}$. Our analysis focuses on the Gaussian mechanism, but the same method can be used with other noise mechanisms.

Let a novel client apply $(\epsilon, \delta)$-DP to the encoder using the Gaussian mechanism. The server sends the encoder $g$ to the client. The client then sends to the server $g(\{x_j^i\}_{j=1}^{m_i}) + \xi$ as its embedding, where $\xi$ is an IID vector from Gaussian distribution with $\sigma^2 = \frac{2(\Delta g)^2 \log(1.25/\delta)}{\epsilon^2}$. To do that, the client must know the sensitivity of the encoder. The following lemma shows that for a DS encoder, we can bound the sensitivity of the encoder, hence bound the noise magnitude necessary to achieve privacy. See proof in Appendix **??**.

**Lemma 2.** *Let $g$ be a deep-set encoder, written as: $g(D) = \psi(\frac{1}{|D|}\sum_{x \in D}\phi(x))$. If $\psi$ is a linear function with Lipschitz constant $L_\psi$, and $\phi$ is bounded by $B_\psi$, then the sensitivity of the encoder is bounded by $\Delta g \leq \frac{2}{|D|}L_\psi B_\phi$.*

The lemma shows that the encoder sensitivity decreases linearly with the size of the novel client dataset $|D|$. For a given $(\epsilon, \delta)$, a lower sensitivity allows us to use less noise to achieve the desired privacy. This in turn means that the client can achieve better performance.

We now empirically evaluate the effect of adding Gaussian additive noise to the embedding of a novel client. To meet the conditions in lemma 2, we normalized the output of $\phi$ to be on a unit sphere, so $B_\phi = 1$. In addition, we average the output of $\phi$, so $L_\psi = 1$. We used $\delta = 0.01$ and compared different values of $\epsilon$ and dataset sizes.

Figure 4 shows that with sufficient data, a novel client can protect its privacy without compromising its performance. For example, given a desired privacy of $\epsilon = 0.3$, if the client feeds the DP-encoder with 3000 samples, the HN creates a personalized model from a perturbed embedding that is as accurate as a non-DP model.

## 8  Conclusion

This paper describes a new real-world FL setup, where a model trained in a FL workflow is transferred to novel clients whose data are not labeled and were not available during training. We describe ODPFL-HN, a novel approach to OD-PFL, based on an encoder that learns a space of clients and an HN that maps clients to corresponding models in an "on-demand" way. We evaluated ODPFL-HN on five benchmark datasets, showing that it generalizes better than current FL and modified PFL methods. We also analyze and bound the generalization error for a novel client and analyze applying DP for the novel client. We hope that this paper will encourage the research community to consider generalization to novel clients when designing FL methods.

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
