# OpenReview forum: "On-Demand Unlabeled Personalized Federated Learning"
_TMLR — Withdrawn by Authors_

### Review · Reviewer_4BsB · 2023-05-05

**Summary Of Contributions:**

The manuscript considers an interesting scenario in personalized FL. It focuses on the deployment scenario and proposes to extend the idea of hyper-network to generate the personalized model for arbitrary clients.

Though the problem of *performing federated learning on novel clients with unlabeled data that are only available at inference time* is interesting and crucial, the current manuscript is far behind in being accepted for publication, due to
* unsounded empirical evaluations,
* limited novelty.


**Audience:**

Yes

**Claims And Evidence:**

No

**Requested Changes:**

1. Add more SOTA baselines for a fair comparison.
2. Also conduct experiments for standard pFL settings.
3. Clarify the significance of section 7, and elaborate on the findings of section 5.


**Strengths And Weaknesses:**

## Strengths
* In general, the paper is well-structured and readers can get the key idea easily. The problem studied in this paper is well-motivated and seems crucial to the community.
* The manuscript examines the proposed method on various benchmarks/datasets, e.g., CIFAR, iNaturalist, Landmarks, and Yahoo Answers.
* The paper also considers the effects caused by distribution shifts.

## Weaknesses
1. The novelty of this manuscript can be doubted. It is a simple extension of the hyper-network idea, and the main contribution might lie in formulating novel unseen and unlabeled clients for personalized federated learning.
2. Missing baselines. Compared to the version the reviewer found 1 year ago, the current manuscript did not include some other recent strong competitors. For example, methods proposed in [1, 2] should also cover the scenarios of novel unlabeled clients (considered in this paper).
3. Though the reviewer understood the focus of this paper, it is still necessary to include some other FL methods that either enable a stronger global model or strong/robust personalized methods e.g., [1, 2, 3, 4]. Otherwise, it is hard to justify that ODPFL-HN should be the right solution for personalized FL.
4. The exact contribution of including differential privacy in section 7 is unclear to me, due to the missing comparison and discussion.
5. The reference links in the Appendix are missing.
6. The section 5 looks trivial to the reviewer. A detailed discussion or insights are encouraged here.

## References
[1] Test-Time Robust Personalization for Federated Learning. ICLR 2023.

[2] Personalized Federated Learning through Local Memorization. ICML 2022.

[3] Ditto: Fair and Robust Federated Learning Through Personalization. ICML 2021.

[4] On Bridging Generic and Personalized Federated Learning for Image Classification. ICLR 2022.

---

### Review · Reviewer_2yZP · 2023-05-22

**Summary Of Contributions:**

This paper introduces a method called On-Demand Unlabeled Personalized Federated Learning (OD-PFL), which addresses the challenge of applying a FL model trained on a set of clients to novel unlabeled clients at inference time. The proposed approach, ODPFL-HN, learns to produce a new model for the late-to-the-party client by training an encoder network that learns a representation for a client given its unlabeled data and feeding that client representation to a hypernetwork that generates a personalized model for that client. The paper also presents a generalization bound based on multitask learning and domain adaptation, and analysis of differential privacy for a novel client, as well as evaluation on five benchmark datasets showing that ODPFL-HN performs better than or equal to the baselines.

**Audience:**

Yes

**Broader Impact Concerns:**

No concerns on the potential ethical issue.

**Claims And Evidence:**

Yes

**Requested Changes:**

Please address the comments in weaknesses.

Minors: The reference to the appendix is missing.

**Strengths And Weaknesses:**

Strength(s):
1. The paper is clearly written, figures are well designed and the overall flow is easy to follow.
2. This discussion for differential privacy is helpful to understand the potential privacy risk.

Weaknesses:

1. The proposed approach requires unlabeled data from the novel client, which may not always be available in real-world scenarios, especially for a big amount. From figure 4, it seems the number of data samples matter.
2. The technical contribution over [1] is not clear. Although the problem setting is different from [1], the overall methods look similar (please correct me if I am wrong). If this is the case, to highlight the contributions, the authors are suggested to show the connection to [1] and elaborate on the technical difficulties of adapting [1] to the problem setting discussed in this paper.
3. The problem theoretical analysis (generalization bounds and DP) are limited to IID assumptions.
4. The paper does not provide a detailed comparison with other state-of-the-art methods. The author should discuss and compare with [2] which seems adaptable to the setting this discussed in this paper. More recent PFL methods, such as  Ditto, MOON, should be included for fair comparison. Also, some test-time methods can be adapted to FL (please refer to the ones used in [2]).
5. The reported results in this manuscript are much lower than the ones reported in the original papers of the baselines (e.g., FedMA [33]). Please provide justification for the differences.
6. The paper does not explicitly discuss the computational complexity of the proposed approach and it is helpful to report them.

References:

[1] Aviv Shamsian, Aviv Navon, Ethan Fetaya, and Gal Chechik. Personalized federated learning using
hypernetworks. In International Conference on Machine Learning, pp. 9489–9502. PMLR, 2021.

[2] Jiang, L. and Lin, T., 2022. Test-Time Robust Personalization for Federated Learning. ICLR 2023.

---

### Review · Reviewer_TPte · 2023-05-26

**Summary Of Contributions:**

This paper studies a novel setting in federated learning, where some clients want to use the federated learning model but they do not have labels such that one cannot fine-tune the global model on their local data. The vanilla single global model trained by FL algorithm may not work well for these new clients, as their data may have different distributions from the training clients. In order to address this challenge, the authors proposed a method based on hyperNet (HN). Specifically, each new client first inputs its local data into an encoder, which output a descriptor (or kind of high-level representation) of the local data. Then, the descriptor will be used as the input for the hyperNet to get a personalized model. Extensive experiments on several different datasets validate the effectiveness of the proposed method.

**Audience:**

Yes

**Broader Impact Concerns:**

Since the server knows both the architecture and outputs of the client encoder, it may be easy to recover the client input data.

**Claims And Evidence:**

Yes

**Requested Changes:**

The authors should address my concerns on practicality and privacy of the proposed method.

**Strengths And Weaknesses:**

Strengths
1. The studied problem is quite new in federated learning community. I feel this is a valid problem that may be very common in practice. This paper may have large impacts.
2. The experiments are thorough. The authors evaluated their methods on >4 datasets and observed significant performance benefits of using the proposed method.


Weakness
1. I have a big concern on the practicality of the proposed method. In particular, the client needs to send something and receive a personalized model back. This means, the server needs to save the identity of each client. This may violate the federated learning's requirement that the server cannot identify and distinguish its clients. As a result, it is not compatible with secure aggregation protocol. Also, if there are a large amount of new clients, then the server's compute and memory loads will become prohibitive.
2. Another concern is about the privacy of this method. Although the authors state that we can use differential privacy to protect the sample-level privacy on each client, what is more important in federated learning is the user-level privacy, which cannot be guaranteed in this method. In addition, since the server knows both the architecture and the outputs of the client encoder, it may be very easy to recover clients input data.

---

### Comment · Action_Editors · 2023-04-18
**About the appendix**

Dear authors,

I saw "See Appendix ?? for a detailed proof" on Page 6 since you uploaded the appendix as a separate pdf file. Please fix this minor issue in your revision.

AE

---

### Comment · Action_Editors · 2023-04-26
**About the assignment acknowledgement**

Hi reviewers Zaad and TPte,

Can you please complete the task of **assignment acknowledgement** when you have time? Thanks!

AE

---

### Comment · Action_Editors · 2023-05-19
**Reminder**

Dear reviewers (except 4BsB),

Hope you are doing well and good luck with your NeurIPS submissions! Since the NeurIPS deadline has already passed, I wonder when you can start reviewing this paper. Can you acknowledge that you have seen this reminder?

AE

---

### Comment · Action_Editors · 2023-06-08
**Author feedback reminder**

Dear authors,

The rebuttal period is 2 weeks, started 13 days ago, and thus is about to end in 1 or 2 days. Please address the reviewers' concerns, if you would like to do so, as soon as possible.

AE

---

### Note · Authors · 2023-06-22

I have read and agree with the venue's withdrawal policy on behalf of myself and my co-authors.